# A capacitive laser-induced graphene based aptasensor for SARS-CoV-2 detection in human saliva

Geisianny Moreira[1,2,3], Hanyu Qian[4], Shoumen Palit Austin Datta[5,6], Nikolay Bliznyuk[4], Jeremiah Carpenter[7,8], Delphine Dean[7,8], Eric McLamore[1,2,3]*, Diana Vanegas[1,3,9]*

1 Environmental Engineering and Earth Sciences, Clemson University, Clemson, South Carolina, United States of America, 2 Department of Agricultural Sciences, Clemson University, Clemson, South Carolina, United States of America, 3 Global Alliance for Rapid Diagnostics, Michigan State University, East Lancing, Michigan, United States of America, 4 Department of Agricultural and Biological Engineering, University of Florida, Gainesville, Florida, United States of America, 5 Department of Mechanical Engineering, MIT Auto-ID Labs, Massachusetts Institute of Technology, Cambridge, Massachusetts, United States of America, 6 Department of Anesthesiology, Medical Device (MDPnP) Interoperability and Cybersecurity Labs, Biomedical Engineering Program, Massachusetts General Hospital, Harvard Medical School, Cambridge, Massachusetts, United States of America, 7 Center for Innovative Medical Devices and Sensors (REDDI Lab), Clemson University, Clemson, South Carolina, United States of America, 8 Department of Bioengineering, Clemson University, Clemson, South Carolina, United States of America, 9 Interdisciplinary Group for Biotechnology Innovation and Ecosocial Change -BioNovo, Universidad del Valle, Cali, Colombia

* dvanega@clemson.edu (DV); emclamo@clemson.edu (EM)

**Data Availability Statement:** All relevant data are within the manuscript, Supporting information files, and are available through the NIH RADx-rad

## Abstract

SARS-CoV-2 virus induced CoVID-19 pandemic has accelerated the development of diagnostic tools. Devices integrated with electrochemical biosensors may be an interesting alternative to respond to the high demand for testing, particularly in contexts where access to standard detection technologies is lacking. Aptamers as recognition elements are useful due to their stability, specificity, and sensitivity to binding target molecules. We have developed a non-invasive electrochemical aptamer-based biosensor targeting SARS-CoV-2 in human saliva. The aptamer is expected to detect the Spike protein of SARS-CoV-2 wildtype and its variants. Laser-induced graphene (LIG) electrodes coated with platinum nanoparticles were biofunctionalized with a biotin-tagged aptamer. Electrochemical Impedance Spectroscopy (EIS) for BA.1 sensing was conducted in sodium chloride/sodium bicarbonate solution supplemented with pooled saliva. To estimate sensing performance, the aptasensor was tested with contrived samples of UV-attenuated virions from 10 to 10,000 copies/ml. Selectivity was assessed by exposing the aptasensor to non-targeted viruses (hCoV-OC43, Influenza A, and RSV-A). EIS data outputs were further used to select a suitable response variable and cutoff frequency. Capacitance increases in response to the gradual loading of the attenuated BA.1. The aptasensor was sensitive and specific for BA.1 at a lower viral load (10–100 copies/ml) and was capable of discriminating between negative and positive contrived samples (with strain specificity against other viruses: OC43, Influenza A, and RSV-A). The aptasensor detected SARS-CoV-2 with an estimated LOD of 1790 copies/ml in contrived samples. In human clinical samples, the aptasensor presents an accuracy of 72%, with 75% of positive percent of agreement and 67% of negative percent of

open-source data hub (https://www.radxdatahub. info/).

**Funding:** Research reported in this publication was supported by the National Institute on Alcohol Abuse and Alcoholism of the National Institutes of Health under Award Number U01AA029328 (DV). The content is solely the responsibility of the authors and does not necessarily represent the official views of the National Institutes of Health. Additional support was provided by the COVID Research Seed Grant Program, Clemson University (Award #2021001151), and the Startup Funding from the Department of Environmental Engineering and Earth Sciences of Clemson University (DV). NIH P20GM121342 supported sequencing efforts (DD). The funders had no role in study design, data collection and analysis, decision to publish, or preparation of the manuscript.

**Competing interests:** The authors have declared that no competing interests exist.

agreement. Our results show that the aptasensor is a promising candidate to detect SARS-CoV-2 during early stages of infection when virion concentrations are low, which may be useful for preventing the asymptomatic spread of CoVID-19.

## Introduction

Development of devices for rapid diagnosis of diseases has accelerated in response to the recent CoVID-19 pandemic and the potential emergence of new pathogens [1–3]. Performing rapid diagnostics in a pandemic context requires devices with high analytical specificity and sensitivity, simple and rapid testing workflows, and amenability to expedited manufacturing for wide deployment [2]. Electrochemical biosensors are an exciting alternative for responding to periods of high testing demand, particularly in contexts where access to clinical infrastructure and standard laboratory analyses is lacking [4]. Electrochemical biosensors operate upon interfacial electroanalysis of supramolecular events (e.g., antigen-receptor coupling, enzymatic reaction catalysis, etc.) that occur between an immobilized biorecognition element, and a target analyte present in the sample. Electrochemical biosensors can be classified into two subclasses, namely metabolism and affinity biosensors. Operation principle of each subclass is defined by the type of interaction that will generate a recognition signal [5]. Various molecules can be used as recognition elements in electrochemical biosensors, such as enzymes, antibodies, peptides, lectins, carbohydrates, and aptamers [6].

Electrochemical aptamer-based (E-AB) biosensors have been recently implemented for disease diagnosis [7]. Compared to other biomolecules, aptamers stand out due to analytical competitiveness from their inherently high specificity and sensitivity toward the target analyte [8]. Aptamers are engineered single-stranded DNA/RNA sequences (typically composed of 20–60 nucleotides) that can attain antigen specificity comparable, and in some instances superior, to antibodies [7, 9]. Aptamers can be quickly obtained using Systematic Evolution of Ligands by Exponential Enrichment (SELEX), offering an additional advantage for the rapid development of capture probes aimed at detecting emerging viruses with high mutation capacity. For example, SARS-CoV-2, in less than 3 years, has already evolved into 31 variants [10, 11]. To date, Omicron is classified as a variant of concern with 21 subvariants [12].

Detection of intact SARS-CoV-2 in biofluids (e.g., saliva) can be achieved by targeting exposed structural proteins on the virion particle, such as the spike protein (S), which is abundant and physically accessible [13]. A few aptamers have been developed to bind SARS-CoV-2 with high affinity independent of mutations in the Spike protein [14–16]. In general, aptamers require specific conditions (e.g., buffer composition, ions, pH) to bind the target analyte [17, 18]. These requirements make the reproducibility of binding events low when the ultimate goal of the aptasensor is detection in biofluids, for example. Therefore, applying aptamers as the recognition element in electrochemical devices using a simple saline solution as a binding buffer has yet to be explored.

Graphene-based electrochemical devices have improved with the development of laser-induced graphene (LIG) electrodes by Tour group [19]. LIG approach are one-step and scalable process to fabricate electrodes with high electrical conductivity using cost-affordable materials [19]. A relevant aspect of biomedical application of electrochemical sensing is metadata analysis for accurate diagnosis. Electrochemical techniques may generate data with several potential response variables. Data analytic tools based on statistical machine learning (SML) may be essential to select a suitable response variable and correct interpreted biosensors output. SML can provide rapid and easy-to-use tools to score, process, and assign risk for biosensor-based diagnosis devices [20].

Herein, we develop a non-invasive aptasensor targeting SARS-CoV-2 in human saliva. Laser-induced graphene (LIG) electrodes were used as transducer surfaces. Electrochemical impedance spectroscopy (EIS) was applied as signal transduction for SARS-CoV-2 sensing. Bio-layer interferometry approach was applied for a qualitative screening of aptamer binding ability towards the whole attenuated viral particle. Data analytic tool based on statistical machine learning was applied to analyze aptasensor output. Finally, we used clinical samples as a proof-of-concept of aptasensor SARS-CoV-2 detection in human saliva.

## Materials and methods

### Chemicals and biomaterials

Potassium ferrocyanide [$K_4Fe(CN)_6$], potassium ferricyanide [$K_3Fe(CN)_6$], and Tween 20™ were obtained from Thermo Fisher Scientific (Waltham, MA, United States). Potassium chloride (KCl), chloroplatinic acid solution ($H_2PtCl_6$), and lead (II) acetate trihydrate [$Pb(CH_3CO_2)_2 \cdot 3H_2O$] were purchased from Sigma Aldrich (St. Louis, MO, United States). Kapton film (electrical grade polyimide film, 0.005" thick) was obtained from McMaster-Carr (Elmhurst, IL, United States). Sodium chloride/sodium bicarbonate ($NaCl/NaHCO_3$) physiological home-use solution (Sinus Rinse™, NeilMed) was obtained from a local pharmacy. Pooled saliva (Pooled human donors, pre-COVID, Catalog number: 991-05-P-PreC-5) was obtained from Lee Biosolutions, Inc. (Maryland Heights, MO, United States). Octet® Streptavidin (SA) biosensors were purchased from Sartorius AG (Germany), and black 96-well flat-bottom microplates (Greiner Bio-One) were purchased from Fisher Scientific (Waltham, MA, United States).

DNA aptamers were synthesized by GeneLink, Inc. (Orlando, FL, United States) with a 5'modification to insert a Biotin-TEG. Recombinant streptavidin (N-terminal 6X His-tag; *Escherichia coli* expression system; 18 kDa; >95% purity, Catalog number: ab78833) was purchased from Abcam, Inc. (Cambridge, United Kingdom). Recombinant spike protein RBD domain (C-terminal 6X His-tag; Sf9 insect cells expression system; 25 kDa; >90% purity; Catalog number: Z03479) was obtained from Genscript, Inc. (Piscataway, NJ, United States). Attenuated SARS-CoV-2 virus samples were obtained from the University of California San Diego through the NIH RADx-rad Diagnostics Core Center (DCC).

### Aptamer selection

A literature review was performed to select aptamers designed for SARS-CoV-2 detection. Selection was based on the following features *i)* binding affinity; *ii)* binding buffer composition; *iii)* molecular target; *iv)* binding ability towards variants of concern; *v)* binding assays towards intact viral particles. In silico modeling was performed for aptamer structure using RNAfold [21] and Mfold [22] web servers. In silico modeling on MFold was performed with the following parameters using DNA folding form: DNA sequence linear, folding temperature of 37 ˚C, $Na^+$ 1M and $Mg^{++}$ 1M for ionic conditions. In silico modeling on RNAfold was performed using the minimum free energy (MFE), partition function algorithm, and DNA parameters [23] with a folding temperature of 37 ˚C.

### Sensor fabrication

Our proposed aptasensor was fabricated using laser-induced graphene (LIG) electrodes [19]. LIG-single working electrodes were engraved on a Kapton film substrate using a $CO_2$ laser (VLS2.30DT, Universal Laser Systems, Inc., Scottsdale, AZ, US) at 75% speed, 40% power, and 1000 PPI. The working electrode was composed of a circular working area (3.0 mm) connected

to a stem (14.3 x 2 mm) that leads to a bonding pad area (2.9 x 2.5 mm). A passivation layer of nitrocellulose lacquer was applied to the stem area, and the metallic tape was incorporated into the bonding pad area of the working electrode. LIG-single working electrodes were used in experiments with UV-attenuated viruses. We used a LIG three-electrode (LIG-chip) system to test human clinical samples with active SARS-CoV-2 infection. LIG-chip includes a reference electrode, an auxiliary electrode, and a working electrode as a whole system. LIG-chip was engraved on a Kapton film substrate using the same settings previously described for LIG single-working electrode. A metallic tape was incorporated into the bonding pad area of all three LIG electrodes. A nitrocellulose passivation layer was applied to the stem area of all electrodes. A reference electrode was developed with metal adhesive tape (nickel alloy). A counter (auxiliary) electrode is a bare LIG. The LIG-chip system was adhered to a chemical-resistant PVC (11.7mm) as physical support using double-sided tape. Both LIG-single and LIG-chip were designed in CorelDraw. Fig 1 shows a schematic figure of LIG-single and LIG-chip. Platinum nanoparticles (nPt) were incorporated on the working electrode for both electrodes systems (LIG-single and LIG-chip) via electrodeposition in a solution of 1.44% (v/v) chloroplatinic acid and 0.002% (v/v) lead acetate in a constant potential of 10 V for 90s, according to the method described in Moreira et al. [24].

Cyclic voltammetry technique was applied to select platinized electrodes with a similar electrochemical response to compose triplicates. CV testing was carried out in a solution of 100 mM KCl, 2.5 mM $K_3[Fe(CN)_6]$, and 2.5 mM $K_4[Fe(CN)_6]$ with a potential range from −0.8 to 0.8 V at a scan rate of 200 mV/s for 10 cycles. A benchtop MultiPalmSens4 potentiostat (PalmSens, Houten, Netherlands) was used in the electrochemical CV testing. For LIG-single selection, a commercial (BASi®, West Lafayette, IN, United States) Ag/AgCl (3M KCl) electrode was used as a reference electrode, and a platinum wire (7.5 cm) as an auxiliary electrode. The similarity between voltammogram shapes and overlap of oxidation and reduction peaks were used as visual parameters to select the triplicates for sensing studies.

## Determination of saliva potential by Open Circuit Potential technique

Open Circuit Potential (OCP) technique was applied to determine saliva's potential (V) and different interferent substances that could mislead the results when human saliva is not used

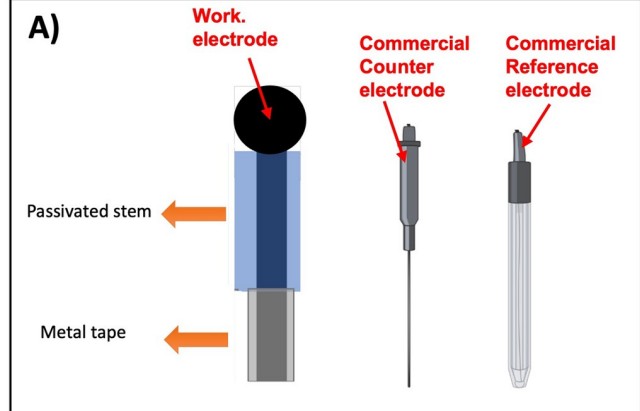
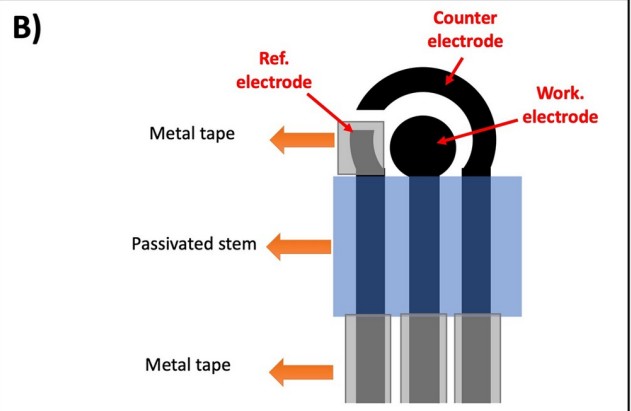

**Fig 1. Schematic representation of laser-induced graphene electrodes.** (A) LIG-single working electrode. (B) Three electrode system—LIG-chip. In both systems, the working electrode was composed of a circular working area (3.0 mm) connected to a stem (14.3 x 2 mm) that leads to a bonding pad area (2.9 x 2.5 mm).

in the sensing. LIG-single bare (without platinum nanoparticles) working electrodes were fabricated as previously described. The following solutions were selected for testing: human saliva, pooled saliva, artificial saliva, tap water, orange juice, Gatorade, coffee, synthetic urine, and NaCl/NaHCO$_3$ (212.8 mmol/L of NaCl and 64.4 mmol/L of NaHCO$_3$) buffer. Solutions were diluted in NaCl/NaHCO$_3$ buffer to get a concentration of 10% (v/v) of each solution. OCP testing was performed by immersing LIG-single bare working electrodes in 10% of each solution, using a t interval of 0.1s for 60 seconds.

## Sensor biofunctionalization

LIG-nPt (single and chip) electrodes were biofunctionalized with a biotin-tagged aptamer via streptavidin-biotin coupling method. First, working electrodes were polarized via chronoamperometry at +1.0 V for 600s to facilitate the immobilization of Streptavidin, which due to its net negative surface charge, allows electrostatic interactions with the working electrode net positive surface charge. Second, 10 μl of a streptavidin his-tagged suspension (1 mg/ml) was drop-casted into the working electrode surface, incubated for 10 minutes at room temperature, and rinsed with a buffer solution before measurement. Third, the streptavidin-coated working electrodes were drop-casted with 10 μl of an aptamer suspension (1 mg/ml), incubated for 10 minutes at room temperature, and rinsed with a buffer solution. Confocal imaging of LIG electrodes before and after biofunctionalization can be found in S5 Fig in S1 File.

Electrochemical Impedance Spectroscopy (EIS) testing was performed for each step of the biofunctionalization process (bare working electrode, his-tagged streptavidin loading, and aptamer loading) to evaluate changes in electrochemical signal in response to adding bio layers into the working electrode surface. Electrochemical measurements were performed in a three-electrode cell stand connected to a benchtop potentiostat (PalmSens®, MultiPalmSens4, Houten, Netherlands) as previously described. EIS testing was carried out in a non-Faradaic mode by immersing the biofunctionalized electrodes in a physiological solution (212.8 mmol/L of NaCl + 64.4 mmol/L of NaHCO$_3$) supplemented with 10% (v/v) pooled saliva, pH 8, at 22˚C and 1 atm. EIS settings for all testing were: frequency range of 0.01 Hz–10,000 Hz, AC amplitude of 0.08 V, and a DC voltage of 0.36 V. EIS tests yield data on total impedance (Z), real impedance (Z′), imaginary impedance (Z″), series capacitance (Cs), real capacitance (C′), and imaginary capacitance (C″) at 63 cutoff frequencies within the described frequency range. Biofunctionalized electrodes (a.k.a, Aptasensors) were used for electrochemical detection studies against attenuated SARS-CoV-2 and non-target respiratory viruses.

## Attenuated SARS-CoV-2 BA.1 sensing and benchmarking with non-target viruses

Titration experiments were conducted to evaluate the EIS response of the aptasensor to clinically relevant concentration levels of SARS-CoV-2 in saliva [13]. The SARS-CoV-2 Omicron BA.1 UV-attenuated virus was spiked in a physiological solution (212.8 mmol/L of NaCl and 64.4 mmol/L of NaHCO$_3$) supplemented with 10% (v/v) pooled saliva to obtain the concentrations of 0, 100, 1000 and 10,000 copies of ORF1a/ml. Then, each concentration solution was drop-casted onto aptasensor surface and incubated for 10 min at room temperature. After incubation, aptasensors were rinsed and then tested using EIS. All EIS testing was conducted in a non-Faradaic mode by immersing the aptasensors in the physiological solution with the same conditions and settings previously described.

The selectivity of the aptasensor towards SARS-CoV-2 was evaluated by benchmarking the aptasensor against UV-attenuated seasonal viruses, such as Common-cold (hCoV-OC43), Cold-like virus (Respiratory Syncytial Virus, RSV-A), and Influenza A virus (H3N2).

Aptasensors were incubated with different concentrations from 10 to 10,000 copies/ml of each virus spiked in a physiological solution (NaCl/NaHCO$_3$) supplemented with 10% (v/v) pooled saliva. EIS testing was carried out under the same conditions as the sensing experiments.

## Data analysis

Potential (V) was evaluated as a response variable from OCP analysis to determine saliva potential. The potential average (*n = 3*) was compared between solution types by statistical analysis (one-way ANOVA and Tukey's test). From EIS outputs, complex impedance and capacitance were used as a response variable for sensing experiments. Analyses of the phase diagram determined a range of cutoff frequencies from 0.01 to 0.1 Hz. Qualitative comparisons for selectivity testing were conducted using one-way ANOVA, and Tukey's test was applied to separate significantly different means ($p < 0.05$). Performance metrics (limit of blank and limit of detection) of aptasensor towards attenuated SARS-CoV-2 Omicron BA.1 variant were calculated. Imaginary capacitance (Net ΔC") at 11 cutoff frequencies (ω = 0.01–0.1 Hz) was normalized by baseline subtraction to remove the biosensor's variation. A non-linear model (MnMolecular1) was fitted to obtain the calibration curve. The standard 3σ method [25] was applied to obtain the limit of detection. Analytical sensitivity was calculated as the slope of the calibration curve in the dynamic concentration range where the behavior is approximately linear. All the data analysis was performed in the OriginLab software.

Principal component analysis (PCA) was used to reduce the dimensionality of data by transforming the raw data into a new coordinate system. The inputs of PCA were the selected biosensor features, including the raw and normalized imaginary impedance (Z") and imaginary capacitance (C") measured at 11 cutoff frequencies (ω = 0.01–0.1 Hz). These biosensor features were selected because of their large signal powers and ability to distinguish the SARS-CoV-2 samples from negative control samples effectively. PCA analysis was performed in R software using the dplyr package [26] to reorganize the data and ggplot2 package [27] to make the graph.

## Qualitative screening of aptamers towards viruses by bio-layer interferometry

An OctetRED96 instrument (Sartorius, Inc.) was used for the qualitative binding screening of biotinylated aptamer towards UV-attenuated SARS-CoV-2 virions (Wildtype, Delta, Beta, Alpha, Omicron BA.1), other coronaviruses (hCoV-OC43, hCoV-229E), and seasonal respiratory viruses (RSV-A, RSV-B, Influenza A, Influenza B). Binding assays were performed at 30°C and 1,000 rpm sample agitation, using the following binding buffer: Sodium (38.3 g/L) and sodium bicarbonate (11.6 g/L) physiological solution (NaCl/NaHCO$_3$) supplemented with 10% (v/v) stabilized artificial saliva and 0.01% (v/v) Tween 20™ (pH 7). Experimental steps included hydration of streptavidin (SA) coated biosensors in a binding buffer for 600s; baseline measurement in a binding buffer for 60s; loading of biotinylated aptamer at 2.5 μg/ml for 70s; second baseline measurement in a binding buffer for 60s; then an association with a target solution for 300s and disassociation in binding buffer for 300s.

## Proof of concept in a SARS-CoV-2 positive saliva sample

Banked SARS-CoV-2 positive saliva samples were used to test the aptasensor. Samples were collected from approved studies through Clemson and Prisma Health Institutional Review Boards (Clemson IRB #2021–0703, #2021–0445, and Prisma Health IRB # Pro00099491) [28]. Samples were collected during January 2022-February 2022 from participants at Clemson University. Written informed consent was obtained from all the participants (age >18

years) to collect the saliva samples and to bank the samples for SARS-CoV-2 related research. Samples were striped of identifiable information after collection and prior to any analysis or research use. Samples were confirmed to be Omicron (BA.1) by whole genome sequencing [29]. Briefly, RNA was extracted using magnetic beads and then sequenced on an Illumina NextSeq 5050 at the Clemson University Genomics and Bioinformatics Facility. The sequences were analyzed using nf-core/viralrecon. Sequences were uploaded to Gen-Bank and GISAID.

To perform detection by aptasensor, positive saliva samples were collected and tested positive for COVID-19 at Clemson University Research and Education in Disease Diagnosis and Intervention (REDDI) Lab (CLIA # 42D2193465) and used in the biosensing. The concentration of Omicron sample was confirmed by PCR as $1.16 \times 10^6$ copies of N1 gene/ml. The sample was diluted in physiological solution (212.8 mmol/L of NaCl and 64.4 mmol/L of NaHCO$_3$) supplemented with pooled saliva (10%) to get the concentration within aptasensor working range (10, $10^2$, $10^3$, and $10^4$ copies of N1 gene/ml). Saliva target-free (0 copies/ml) was used as a negative control. Each concentration sample was drop-casted onto the aptasensor surface and incubated for 10 min at room temperature. After incubation, the electrodes were rinsed and tested by EIS under the previously described conditions.

## Results and discussion

### *In silico* modeling of aptamer structure

As a recognition element for our biosensor, we selected an aptamer designed by Zhang et al. [14], namely MSA52, which meets our requirements for selection. MSA52 aptamer was designed and selected to bind wild-type S1 domain of Spike protein and non-discriminatively recognizes spike protein of variants. We have included a T-tail and a biotin tag to the SELEX native sequence to facilitate the linker of aptamer to LIG electrode surface by streptavidin-biotin coupling method.

We performed *in silico* modeling for MSA52 aptamer using the native sequence from SELEX selection discovered by Zhang et al. [14]. To explore secondary structure stability, we scrambled the aptamer sequence by substitution of ten nucleotides to thymine (T) from position 6 to 16, and then we performed *in silico* modeling for the scrambled sequence (referred to as T-scrambled). In addition, the scrambled aptamer was used as a control to test different binding affinities towards the target analyte and discrimination between potential non-target viruses. Fig 2 shows one native and one T-scrambled structural predicted as a high probability by the model for MSA52 aptamer (RNAfold, Vienna). Detailed MSA52 aptamer characterization in mfold and RNAfold (Vienna) web servers for SELEX native and scrambled sequences can be found in Section 1 in S1 File.

Aptamer secondary structure is key for the binding affinity, which allows specific interactions with target molecules [30]. Theoretically, substitutions in aptamer sequence change secondary structure, which may lead to decreases in the binding ability. Random substitution of ten nucleotides in MSA52 native sequence decreases the stem/loop ratio from 1.26 to 0.68 in the T-scrambled aptamer. It is known that the presence of secondary structures, such as hairpin loops, increases the bonding strengths of aptamers [31], and is important to the aptamer-target interactions [32]. In addition, GC content decreased from 54% to 48% when we modified the aptamer native sequence. While the GC content gives stability to the secondary structure, the stem-loop is important to stabilize and guarantee the aptamer conformation [33, 34]. In addition, specific binding motifs, such as hairpin loops, play an important role in target binding [35, 36]. We showed by *in-silico* modeling, using RNAfold/Vienna, that substitutions in the native sequence selected by SELEX process may lead to loss of ability to form the

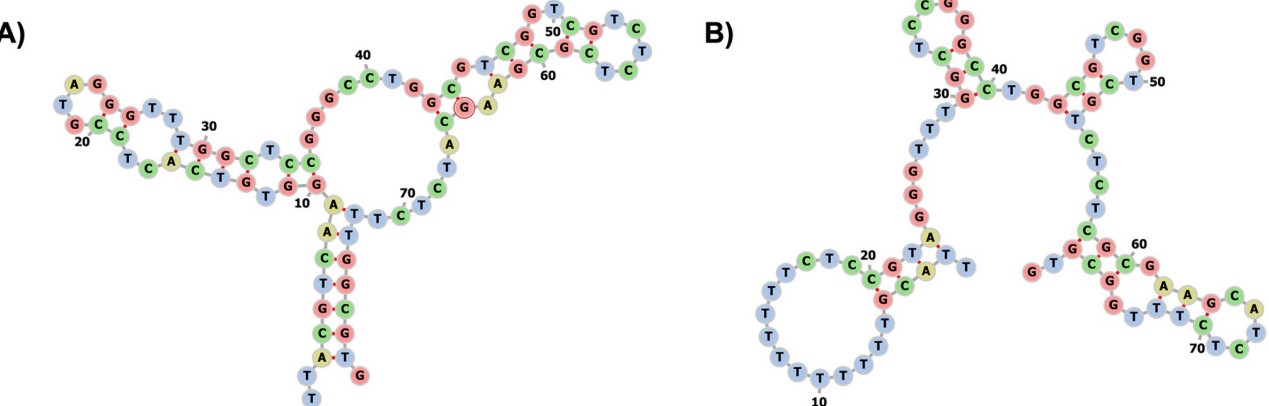

**Fig 2. Predicted secondary structure of aptamer MSA52 by RNAfold (Vienna) web server.** (A) Native sequence from Zhang et al. [14]. (B) T-scrambled sequence (position 6 to 16 modified with T substitution).

secondary structure that is expected to bind the SARS-CoV-2 spike protein, which aptamer was initially designed for Zhang et al. [14].

One of the significant challenges of applying immobilized aptamers is to ensure that these binding motifs maintain their structure and functionality under experimental conditions [37]. A stable secondary structure is essential in electrochemical sensing applications since the aptamer acts as an immobilized recognition element under an applied electric field, which makes *in-silico* modeling a key step in aptasensor development. The secondary structure modeled for MSA52 aptamer native sequence maintain the conservation pattern described by Zhang et al. [14], which plays essential roles in the structure and/or binding function.

## Determination of electrochemical cell constant for various samples

The developed aptasensor was designed for autonomous saliva sampling and later testing in moderate community site settings. Saliva is a complex biofluid matrix and may have potential

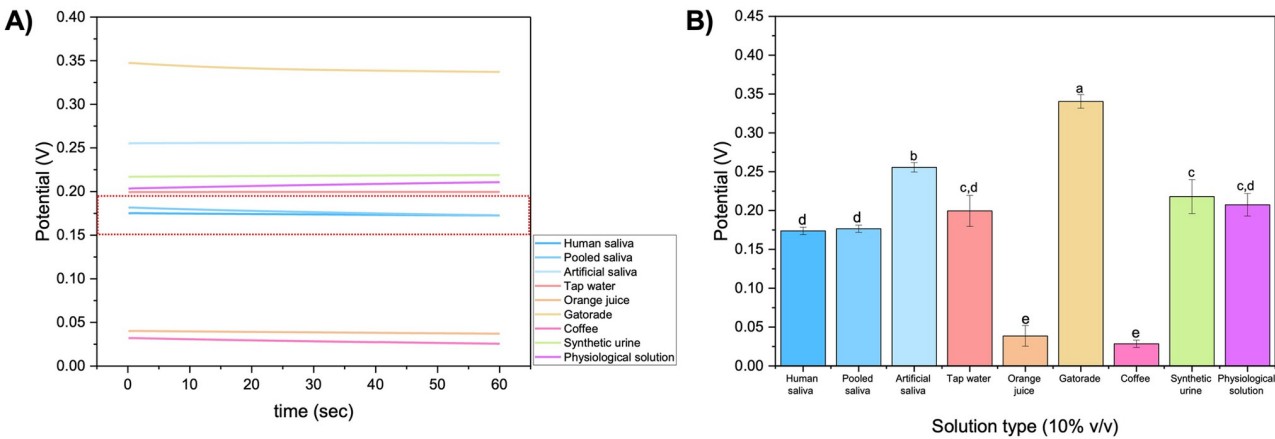

**Fig 3. Determination of electrochemical cell constant ($E_o$) in saliva and various interferent solutions.** (A) A representative OCP graph highlighting the range potential for saliva samples. (B) Average open circuit potential (OCP) for each solution using a 10% sample diluted in physiological solution (NaCl/NaHCO₃). Error bars represent standard deviation from the mean measurement (*n* = 3). Same letters represent groups with no significant difference (p < 0.05).

endogenous/exogenous interference substances. One challenge related to self-handling saliva-collecting is to guarantee that the sample to be tested is human saliva. To guarantee human saliva sample integrity, we used OCP to determine the electrochemical cell constant ($E_o$) of human saliva and potential interference substances that could mislead the test results. A representative OCP output graph is shown in Fig 3A, highlighting that the saliva (human and pooled saliva) potential falls within the range from 0.15 to 0.20 V. Fig 3B shows potential for each solution using a 10% sample diluted in physiological solution (NaCl/NaHCO$_3$). Saliva (human and pooled) potential differs significantly (between 80 to 90%, $p < 0.05$) from potential interferent substances (orange juice, Gatorade, coffee, synthetic urine). The pH measurement of all substances can be found in S4 Table in S1 File.

Electrochemical cell potential can be affected by several factors, such as temperature, concentration, and pressure [38]. Since physical conditions are maintained the same for our experiments, changes in cell potential result from chemical composition of each solution. Cell potential results from redox reactions on the working electrode, which depend on electrolyte conductivity [39], making ions concentration an important factor, as shown by significant differences between solutions such as Gatorade and coffee. It is clear when we see the differences between human saliva (0.17 V) and artificial saliva (0.26 V), showing that this approach can even differentiate between human and artificial saliva ($p < 0.05$). These results demonstrated that with a simple approach, we are able to determine sample integrity before sample testing by our developed aptasensor.

## Aptasensor baseline characterization

A quality control (QC) screening criteria is essential to reduce electrode batch-to-batch variation during LIG electrode fabrication, as Moreira et al. [24] described. For the aptasensor development, we used CV data output (e.g., voltammogram shape and overlapping between oxidation/reduction peaks) as QC screening criteria for triplicates selection (see representative voltammogram in S3 Fig in S1 File). Using a QC criteria for electrochemical studies is important to reduce batch-to-batch variation and guarantee a similar electrochemical behavior during sensing experiments. We applied cyclic voltammetry (Faradaic) at 200 mV/s as a quality control (QC) for selecting replicates of LIG-nPt electrodes for sensing experiments. Representative voltammograms in ferri/ferrocyanide for selected electrode triplicates from a large fabrication batch ($n = 36$) can be found in the S3 Fig in S1 File. Using the QC criteria described here, four to six triplicates are selected from a batch of 36 LIG-nPt electrodes. After selecting electrodes using the QC process, triplicate electrodes were biofunctionalized with his-tagged streptavidin and biotin-tagged MSA52 aptamer.

We measured each loading step using EIS technique. We analyzed different EIS output data to select the most suitable response variable and cutoff frequencies. Negative phase (-φ), total impedance (Z), real and imaginary impedance (Z' and Z"), real and imaginary capacitance (C' and C") was used to characterize aptasensor baseline and select cutoff frequencies. The phase diagram (log $f$ vs. -φ) shows that LIG bare (electrode baseline) has a stable negative phase at a lower frequency range (0.01 to 0.1 Hz) (Fig 4A). In addition, the negative phase for loading steps varies from 70 (LIG-nPt bare) to 55 (biotin-tagged MSA52 aptamer), indicating the capacitative behavior of the aptasensor. Fig 4B and 4C show imaginary impedance and imaginary capacitance plots vs. log of frequency (0.01 to 10,000 Hz). Due to the capacitor behavior of our aptasensor, imaginary capacitance shows distinguish differences for each loading step at a lower frequency range (0.01 to 0.1 Hz). Finally, Nyquist capacitive plot (C' vs. C") in Fig 4B show an evident signal change in response to addition of streptavidin layer on working electrode followed by the addition of a biotinylated MSA52 aptamer layer. Representatives plot for

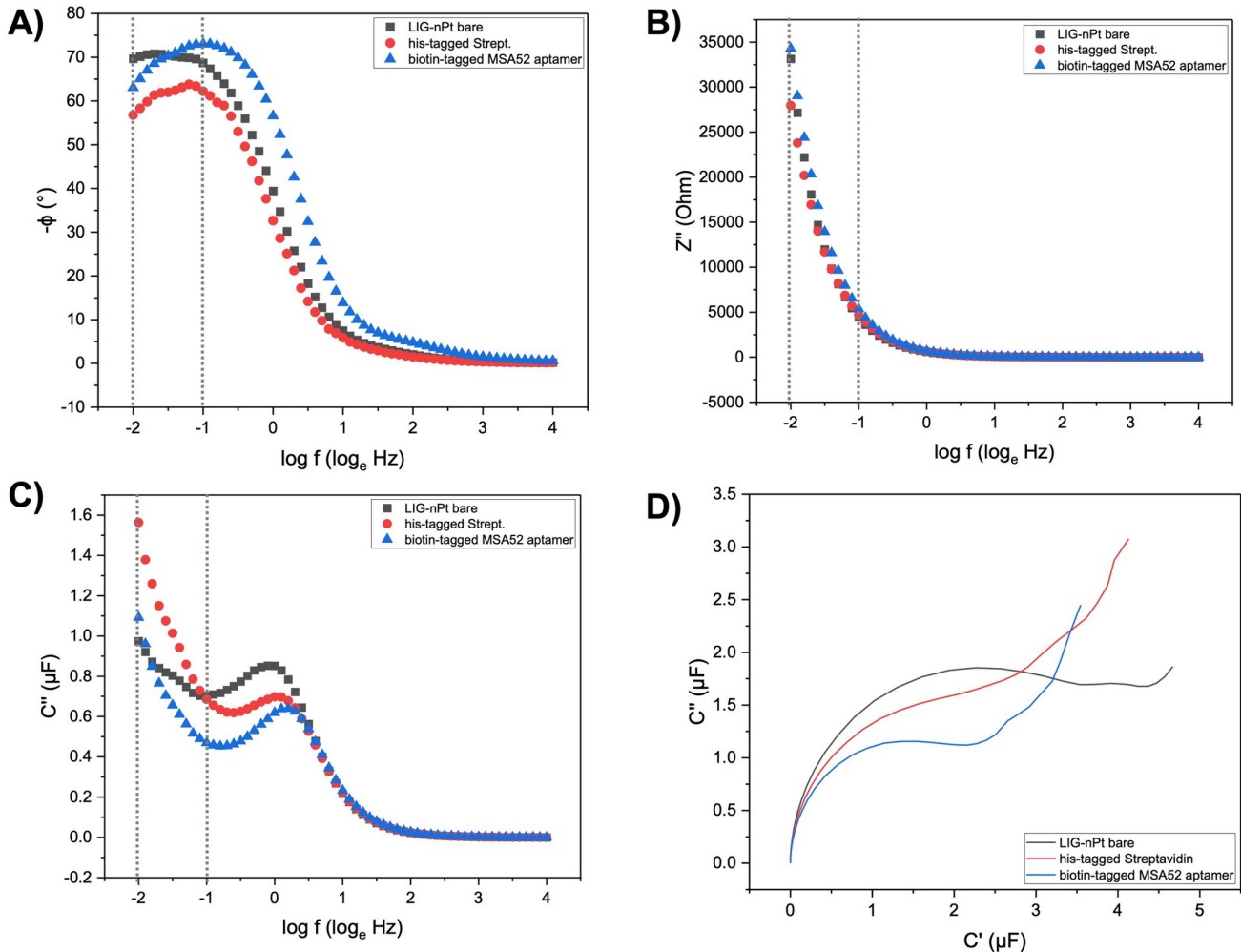

**Fig 4. Electrochemical aptamer-based baseline characterization.** (A) Representative Phase diagram (log *f* vs. -φ) highlighting the selected cutoff frequency range (0.01 to 0.1 Hz) in EIS non-Faradaic mode (physiological solution (NaCl/NaHCO₃) supplemented with 10% (v/v) pooled saliva). (B) Representative Bode Impedance plot (log *f* vs. Z") in EIS non-Faradaic mode (physiological solution (NaCl/NaHCO₃) supplemented with 10% (v/v) pooled saliva). (C) Representative Bode Capacitance plot (log *f* vs. C") in EIS non-Faradaic mode (physiological solution (NaCl/NaHCO₃) supplemented with 10% (v/v) pooled saliva). (D) Nyquist capacitive plot (C′ *vs.* C″) of biofunctionalization with biotin-tagged MSA52 aptamer immobilized on the working electrode *via* the streptavidin-biotin coupling method.

total impedance (Z), series capacitance (Cs), real impedance (Z'), and real capacitance (C') can be found in the Supporting information (S4 Fig in S1 File).

We adopted the streptavidin-biotin coupling method to attach the aptamer to the electrode surface. Several linkers can be used to perform immobilization of aptamers, such as thiol coupling and amine coupling [40]. However, biotin-tagged aptamer doesn't require preparation (reduction to a thiol group) as thiol-tagged aptamers [41]. Also, it provides a successful immobilization by strong bound with a high affinity between streptavidin-biotin ($K_D$ = 1 fM) [42]. Finally, the 5' termini of the aptamer are labeled with a biotin-tag which may allow a correct aptamer conformation during the immobilization. Streptavidin-biotin capture technique is a common approach to immobilize aptamers in assays such as surface plasmon resonance [43]. The advantages of using streptavidin-biotin coupling to immobilize aptamers include *i)* a single-step immobilization which eases capture by streptavidin, *ii)* less constrained movement of the aptamer to avoid steric hindrances during target binding, *iii)* covalently linked to the end

of the aptamer which allows oriented immobilization for conformation into a secondary structure, *iv)* streptavidin has four binding sites for biotin which enhances the attachment of aptamer to the electrode surface [43, 44].

## Electrochemical sensing of attenuated SARS-CoV-2 in contrived samples

Aptasensor specificity toward attenuated SARS-CoV-2 was demonstrated by exposure to Omicron BA.1 (concentration range from 10 to 10,000 copies/ml). A clear sensor response is shown in the calibration curve for UV-attenuated SARS-CoV-2 BA.1 titration (Fig 5A). Capacitive response increases (approximately 150 nF) after exposure to a virus concentration between 10 to 1000 copies/ml. Sensor response is stationary between 1000 to 10,000 copies/ml, implying saturation of aptamer binding sites. The limit of detection ($3\sigma$) was estimated to be 1790 copies/ml using the non-linear response in Fig 5A. The equation used from the non-linear model to estimate LOD can be found in the Supporting information (S5 Table in S1 File). Analytical sensitivity was determined as the slope of the calibration curve in the linear dynamic concentration range (from 10 to 1000 copies/ml) and is estimated as 0.05 µF/copies ml$^{-1}$.

Different from other coronaviruses, such as SARS-CoV and MERS-CoV, SARS-CoV-2 shows a viral load peak (up to 10,000 copies/ml) before the onset of symptoms [45], which facilitates the rapid spread of the virus. A suitable test device should be able to detect the virus

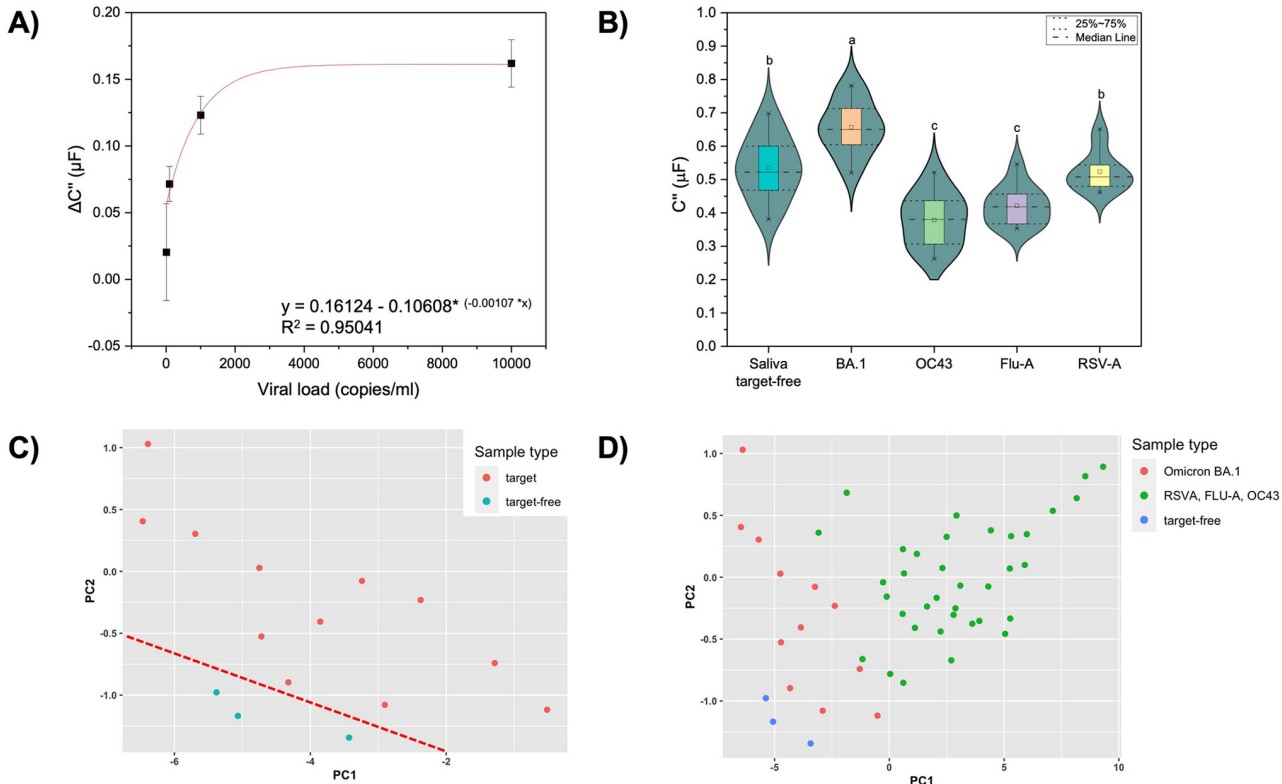

**Fig 5. Electrochemical sensing of SARS-CoV-2 and selectivity performance.** (A) Calibration curve for titration of attenuated SARS-CoV-2 Omicron BA.1. Error bars represent standard deviation from the mean measurement (*n = 12*). (B) Aggregated selectivity for Omicron BA.1 and non-target respiratory viruses (concentration range 10 to 10,000 copies/ml) at a cutoff frequency range from 0.01 to 0.1 Hz (*n = 51*). Same letters represent groups with no significant difference ($p < 0.05$). Scatter plots of first two PCs from PCA: (C) PCA graph for distinguishing between saliva target-free (blue, *n = 3*) and contrived samples (red, *n = 12*). (D) PCA graph for distinguishing saliva target-free (blue, *n = 3*), Omicron BA.1 (red, *n = 12*), and other respiratory viruses (green, *n = 36*).

before symptoms onset to reduce virus spread. Compared with the gold-standard technique for virus detection (a.k.a., Reverse Transcription Polymerase Chain Reaction–RT-PCR), which demonstrate a limit of detection of 1000 copies/mL [46], our aptasensor displays enough sensitivity to perform early detection of SARS-CoV-2. A lower LOD is important in a clinical and epidemiological context since assays with a high LOD will miss more infected patients increasing false-negative rates [47].

The aptasensor was challenged against non-target viruses (hCoV-OC43, RSV-A, Flu-A) in a concentration range of 10 to 10,000 copies/ml. Fig 5B shows a significant (p < 0.05) aptasensor response (C") to target (Omicron BA.1) compared to saliva target-free and non-target viruses. These results demonstrate that the aptasensor displays enough specificity to distinguish SARS-CoV-2 from seasonal respiratory viruses.

To better understand aptasensor response for sensitivity and specificity towards SARS-CoV-2, we apply Principal Components Analysis (PCA) using impedance and capacitance response at 0.01 to 0.1 Hz. Fig 5C and 5D shows the scatter plots of the first two PCs (principal components), which explain 98% of the data variation and capture almost all of the variance in the raw and normalized data. Fig 5C shows the first two PCs for target detection (Omicron BA.1 contrived sample) compared to a target-free sample (negative contrived sample). Fig 5D shows that the aptasensor can discriminate negative contrived samples from positive contrived samples for SARS-CoV-2 BA.1 with strain specificity against other viruses (OC43, FLU-A, and RSV-A). Even though the aptasensor has demonstrated sensitivity and specificity towards the target, a high background signal is detected from the target-free samples due to the saliva being a complex matrix. In addition, non-specific binding is detected for a non-target virus, namely RSV-A. Therefore, even though our detection platform exhibits potential to discriminate between target and non-target viruses, it can be improved to increase its specificity. A suggested approach would be to multiplex the aptasensor with another biorecognition element, for example, an ACE2-based biosensor [24].

## Qualitative screening of MSA52-aptamer towards respiratory viruses

We performed a qualitative screening of MSA52-aptamers (SELEX native and T-scrambled sequences) using bio-layer interferometry technology to evaluate their affinity towards other viruses since electrochemical studies showed a slight response to non-target respiratory viruses. MSA52 aptamer was discovered and described by Zhang et al. [14] but not screened towards negatives controls or non-target viruses in order to evaluate its selectivity towards SARS-CoV-2 viruses. Fig 6A shows an affinity map for native MSA52 towards a high titer of several UV-attenuated respiratory viruses. The $k_{on}/k_{off}$ map shows that native MSA52 is able to bind faster (high $k_{on}$) Influenza B virus than other viruses, such as SARS-CoV-2 variants. However, the complex MSA52-Influenza B unbinds faster (high $k_{off}$) than other viruses, showing weak stability of formed complexes. Although the association constant between MSA52 and SARS-CoV-2 variants is not high, the dissociation between MSA52-SARS-CoV-2 is slower. This demonstrates that the recognition is not strong, but they become stable once the complexes are formed. From a detection perspective, the stability of the complexes is as essential as a rapid recognition between biorecognition molecule and target analyte. Fig 6B shows an affinity map for scrambled MSA52 towards a high titer of several UV-attenuated respiratory viruses. When we scrambled MSA52 aptamer sequence, we changed the binding ability, where the aptamer interacts with the target but doesn't necessarily form complexes ($k_{off}$ near zero). Interestingly, for Omicron and Alpha variants, the complex formed with scrambled MSA52 aptamer appears more stable (low $k_{off}$).

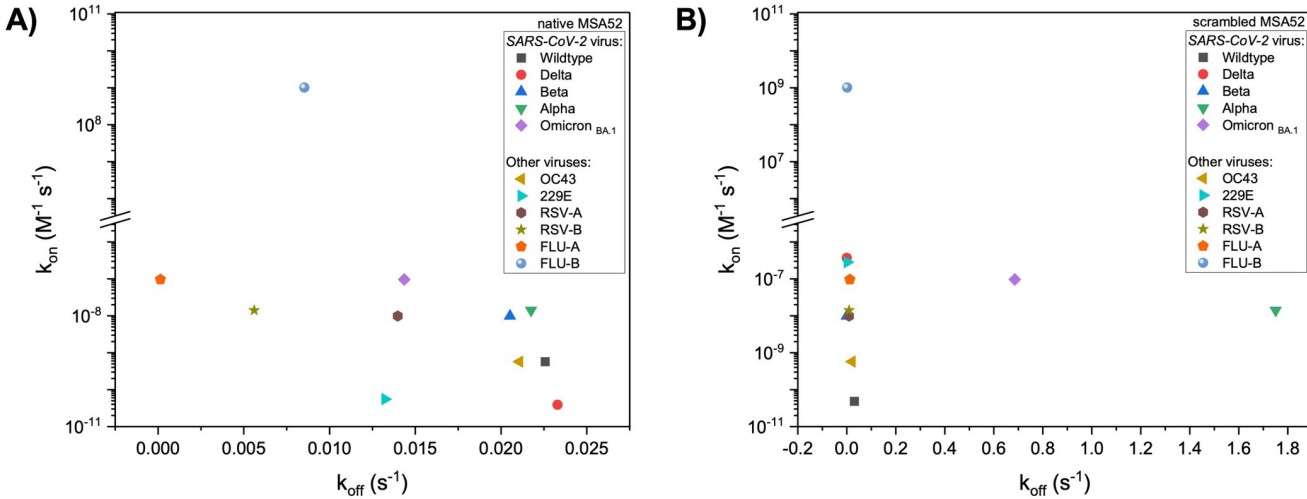

**Fig 6. $k_{on}/k_{off}$ maps for MSA52 aptamer towards respiratory viruses.** (A) Native MSA52 aptamer sequence. (B) T-scrambled aptamer sequence. BLI assay was performed in binding buffer (NaCl/NaHCO$_3$, 10% artificial saliva–mucin, 0.01% Tween20, pH 7.8), with attenuated respiratory viruses, namely: SARS-CoV-2 (Wildtype, Delta, Beta, Alpha, Omicron BA.1), hCoV-OC43, hCoV-229E, RSV-A, RSV-B, Influenza-A, and Influenza-B.

Although native aptamer is expected to show an affinity for the target more than T-scrambled, the SELEX process and the model (RNAfold, see Fig 2) used to predict secondary structure disregards some important factors related to aptamer performance. First, the binding buffer composition (e.g., ion concentration) used in our experiments differs from that used during aptamer development, which may change its affinity for the target. Second, our experiments were conducted with the whole viral particle, whereas the SELEX process uses only the purified target protein. Third, our binding buffer contains a physiological solution (NaCl/NaHCO$_3$) supplemented with saliva (10% v/v) to mimic the conditions for which the aptasensor was designed (saliva testing). Taken together, all these factors may alter the aptamer affinity towards SARS-CoV-2 that was initially described during aptamer selection by Zhang et al. [14].

## Proof-of-concept for active SARS-CoV-2 infection in human saliva samples

We used banked SARS-CoV-2 positive saliva samples to demonstrate the aptasensor ability to detect SARS-CoV-2 in real testing settings. A titration was performed to obtain the concentration within aptasensor working range (10 to 10,000 copies/ml). Fig 7A shows an increase in signal (approximately 100 nF) in response to the increased concentration of up to 100 copies/ml. Concentrations above 100 copies/ml show a decrease in signal (around 170 nF), demonstrated by hook effects on the aptasensor surface. This result shows that the aptasensor ability to detect the viral load in a complex matrix decrease (for concentrations up to 100 copies/ml) compared to buffer conditions. Since our buffer also contains saliva, we attributed the hook effects to the presence of active viral particles and specific saliva composition depending on the sample donors.

Fig 7B shows the results of a contingency table calculation, which considers false negative (FN), false positive (FP), true negative (TN), and true positive (TP) results. A capacitance signal of 0.17 μF was used as a threshold based on capacitance signal for true negative samples ($n = 6$). Based on the established threshold, from 12 true positive samples and six true negative samples, our aptasensor testing yields three false negatives and two false positives. Our

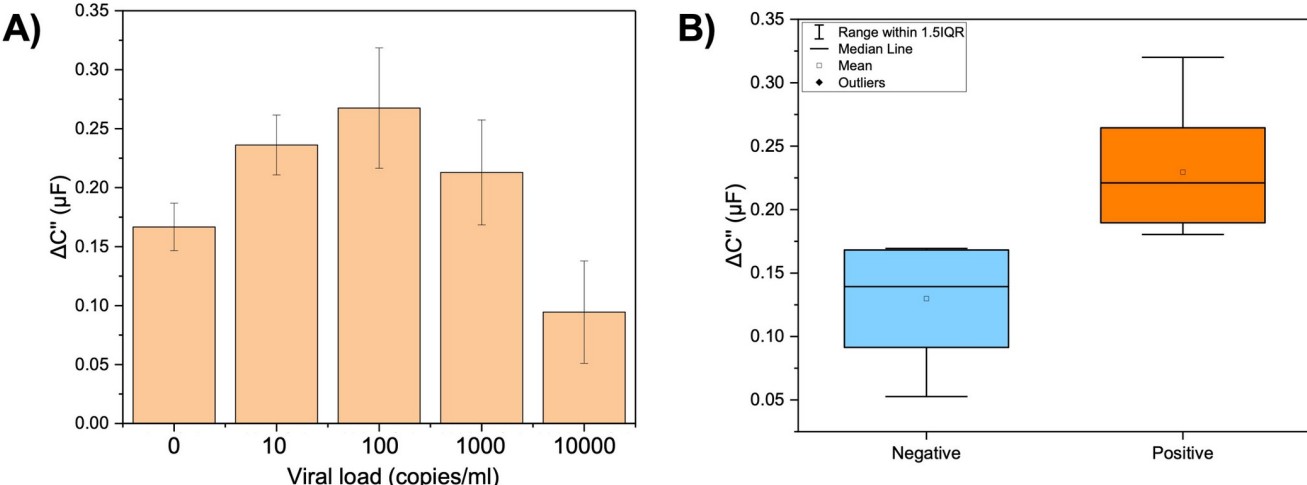

**Fig 7. Proof-of-concept of SARS-CoV-2 detection by aptasensor in clinical human saliva samples.** (A) Titration of active SARS-CoV-2 Omicron sample diluted in physiological solution (212.8 mmol/L of NaCl and 64.4 mmol/L of $NaHCO_3$) supplemented with pooled saliva (10%) in the concentration range of 10, $10^2$, $10^3$, and $10^4$ copies of N1 gene/ml. Error bars represent the standard deviation from the mean measurement ($n = 3$). (B) Test results for aptasensor testing in clinical human saliva samples based on contingency table calculations ($n = 18$).

aptasensor presents an accuracy of 72%, with 75% of positive percent of agreement (PPA, clinical sensitivity), 67% of negative percent of agreement (NPA, clinical specificity), and a Youden index of 0.42.

Current electrochemical aptamer-based sensors for SARS-CoV-2 detection are shown in Table 1. The most common electrochemical transduction adopted for E-AB has been Square Wave Voltammetry (SWV) and Electrochemical Impedance Spectroscopy (EIS). EIS approach is useful for detecting changes in electrical properties resulting from binding events in the working electrode, while SWV measures changes in current with a controlled variation of the potential [48]. Although E-AB in Table 1 shows a lower LOD (femtogram to nanogram), the aptasensors have not been tested with whole viral particles in a complex sample matrix. We estimated a LOD of 1790 copies/ml ($3\sigma$) in contrived samples (attenuated virus in buffer supplemented with pooled saliva) for our aptasensor allowing the virus detection in its clinically relevant range. The aptasensor presented a low rate of false positives and false negatives, suitable accuracy, and clinical sensitivity and specificity for positive Omicron samples. However, due to mutations in the spike protein, a complementary approach using a generalist and a specialist recognition element could be suitable for detecting emerging variants of concern.

**Table 1. Summary of electrochemical aptamer-based biosensors for SARS-CoV-2 detection.**

| Electrochemical approach | Analytical target | Sample matrix | LOD | Reference |
|---|---|---|---|---|
| Square Wave Voltammetry (SWV) | Recombinant S protein—RBD domain | Serum and artificial saliva | NA | [49] |
| Electrochemical Impedance Spectroscopy (EIS) | Recombinant N protein | Buffer and human serum | 0.077 ng mL$^{-1}$ (buffer) 0.16 ng mL$^{-1}$ (human serum) | [50] |
| Square Wave Voltammetry (SWV) | Recombinant S protein | Buffer (PBS), Human saliva and Viral transport medium | 0.03 fg mL$^{-1}$ | [51] |
| Electrochemical Impedance Spectroscopy (EIS) | Whole virus particle | Human saliva | 1790 copies/mL ($3\sigma$) (contrived samples) | This study. |

## Conclusion

We developed a capacitive aptamer-based biosensor for SARS-CoV-2 detection in human saliva. The detection principle is based on the aptamer's ability to bind wild-type S1 domain of Spike protein and non-discriminatively recognize spike protein of variants. The aptasensor is based on laser-induced graphene electrodes, which allow a cost-affordable and simple one-step fabrication in a label-free format. We used EIS as an electrochemical transducer and demonstrated that capacitance at a frequency range from 0.01 to 0.1 Hz is a suitable response variable for SARS-CoV-2 detection. The aptasensor detected SARS-CoV-2 with a LOD (1790 copies/ml in contrived samples) comparable to the gold-standard technique (RT-PCR for CDC assay as a LOD of 1000 copies/ml). These results demonstrate the aptasensor potential for detection of asymptomatic cases or lower viral load cases as well. The aptasensor was able to discriminate between the target (Omicron) and non-target viruses (seasonal respiratory viruses). When tested for clinical samples, the aptasensor shows a higher accuracy (72%) and low false-negative and false-positive rates (75% for PPA and 67% for NPA). Despite the background signal from saliva in the electrochemical sensing (for both contrived and clinical samples), the aptasensor was able to differentiate the capacitance change in response to target (Omicron variant). To improve sensitivity and specificity of the proposed aptasensor, a complementary approach using a generalist recognition element combined with the specialist aptamer could be suitable for detecting emerging variants of concern. Our findings show that our capacitive aptamer-based biosensor may be suitable for non-invasive early detection of SARS-CoV-2.

## Supporting information

**S1 File. Includes figures and tables related to in silico modeling of aptamer structure, electrochemical and microscopy characterization of LIG electrodes and aptasensor, pH and temperature information of samples used in OCP testing, and non-linear model for aptasensor calibration curve.**
(DOCX)

## Acknowledgments

We want to thank the Data Coordination Core and Diagnostics Core of the RADx-rad program at UCSD for supporting our work. We need to acknowledge Dr. Rooksie Noorai for her help with sequencing and bioinformatics.

## Author Contributions

**Conceptualization:** Eric McLamore, Diana Vanegas.

**Data curation:** Hanyu Qian, Nikolay Bliznyuk.

**Formal analysis:** Geisianny Moreira, Hanyu Qian, Nikolay Bliznyuk.

**Funding acquisition:** Eric McLamore, Diana Vanegas.

**Investigation:** Geisianny Moreira.

**Methodology:** Geisianny Moreira, Jeremiah Carpenter.

**Project administration:** Diana Vanegas.

**Resources:** Eric McLamore, Diana Vanegas.

**Supervision:** Shoumen Palit Austin Datta, Nikolay Bliznyuk, Delphine Dean, Eric McLamore, Diana Vanegas.

**Visualization:** Geisianny Moreira, Hanyu Qian.

**Writing – original draft:** Geisianny Moreira.

**Writing – review & editing:** Shoumen Palit Austin Datta, Jeremiah Carpenter, Delphine Dean, Eric McLamore, Diana Vanegas.

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
