## [Decision Letter · Decision Letter 0]

11 Jul 2023

PONE-D-23-18219A capacitive LIG-based aptasensor for SARS-CoV-2 detection in human salivaPLOS ONE

Dear Dr. Vanegas,

Thank you for submitting your manuscript to PLOS ONE. After careful consideration, we feel that it has merit but does not fully meet PLOS ONE’s publication criteria as it currently stands. Therefore, we invite you to submit a revised version of the manuscript that addresses all the points raised during the review process. Please submit your revised manuscript by Aug 25 2023 11:59PM. If you will need more time than this to complete your revisions, please reply to this message or contact the journal office at plosone@plos.org. Please include the following items when submitting your revised manuscript:A rebuttal letter that responds to each point raised by the academic editor and reviewer(s). You should upload this letter as a separate file labeled 'Response to Reviewers'.A marked-up copy of your manuscript that highlights changes made to the original version. You should upload this as a separate file labeled 'Revised Manuscript with Track Changes'.An unmarked version of your revised paper without tracked changes. You should upload this as a separate file labeled 'Manuscript'.

We look forward to receiving your revised manuscript.

Kind regards,

Sabato D'Auria

Academic Editor

PLOS ONE

Journal Requirements:

Reviewers' comments:

Reviewer's Responses to Questions

**Comments to the Author**

1. Is the manuscript technically sound, and do the data support the conclusions?

Reviewer #1: Yes

Reviewer #2: Yes

2. Has the statistical analysis been performed appropriately and rigorously? 

Reviewer #1: I Don't Know

Reviewer #2: Yes

3. Have the authors made all data underlying the findings in their manuscript fully available?

Reviewer #1: Yes

Reviewer #2: Yes

4. Is the manuscript presented in an intelligible fashion and written in standard English?

Reviewer #1: Yes

Reviewer #2: Yes

5. Review Comments to the Author

Reviewer #1: The manuscript by Geisianny Moreira et al. entitled “A capacitive LIG-based aptasensor for SARS-CoV-2 detection in human saliva” describes the development of a non-invasive electrochemical aptamer-based biosensor targeting SARS-CoV-2 in human saliva. The authors conclude that the aptasensor is a promising candidate to detect SARS-CoV-2 SARS-CoV-2 during early stages of infection when virion concentrations are low, it may be useful for preventing the asymptomatic spread of CoVID-19.

I believe that the results obtained do not bring about a significant improvement in the detection method, data in relation to the wild type structure are not presented, I don't know what advantages there are compared to the classic detection method. In my opinion the work should be improved.

I admit this manuscript with major revision.

In particular:

- The introduction is too long, I would like suggest to shorten it

- The resolution of the graphics is very bad

- I would like suggest to insert the figure 1 in the text (line 153)

- In the caption of figure 1 I would eliminate the data already reported in the text (Lines from 286 to 288).

- In the “In silico modeling of aptamer structure” (line 271) paragraph, it is not clear why make substitutions in the sequence if you already know that they do not bring advantages.

- Are the values obtained from the OCP analysis significant? (is the difference significant?)

Reviewer #2: In the manuscript “A capacitive LIG-based aptasensor for SARS-CoV-2” detection in human saliva”, the authors describe the application of laser- induce graphene electrode to develop a non-invasive aptamer-based electrochemical biosensor to detect SARS-CoV-2 in human saliva.

The article I well written and, in my opinion, this work is of interest to the readers of the Journal but I recommend the publication after a minor revision. In particular:

1. In the manuscript the authors follow the sensor's surface functionalization using the impedance signal variation. Have the authors investigated this procedure using additional methods such as atomic force microscopy (AFM)?

2. In the manuscripts, no information about the reproducibility and the long-term stability of the biosensor developed has been provided by the authors. Please, the authors should provide it.

3. How authors have determined the sensitivity of the proposed biosensor?

4. Figure 4° lacks the mathematical equation used by the authors to calculate the biosensor LOD. Please, the authors should provide it.

5. How have the authors selected the interference substances (coffee, Gatorade, synthetic urine, etc)? What is the pH of these samples? The interferent solutions were diluted 10% v/v in buffer, why have the authors selected this value? Have the authors investigated other concentrations? If yes, what is the matrices effect on the virus binding?

6. In the title the acronymous “LIG” should be specify.

6. PLOS authors have the option to publish the peer review history of their article (what does this mean?). If published, this will include your full peer review and any attached files.

Reviewer #1: No

Reviewer #2: No

---

## [Author Response · Author response to Decision Letter 0]

2 Aug 2023

Dear Academic editor and reviewers,

We conducted a point-by-point revision of our manuscript, considering all the reviewers' comments. We are re-submitting an updated version of the manuscript, figures and supplementary material. We would like to thank the reviewers for their comments. The feedback helped to improve the quality of the manuscript and strengthen the ideas that we were trying to convey. Below you can find the answers to each comment. We hope this version of the manuscript satisfies the expectations of your journal. The changes in the paper are highlighted in red. Please let me know if you have any questions.

Reviewer #1

#1 The introduction is too long, I would like suggest to shorten it. 

Thank you for your suggestion. We improved the introduction by removing unneeded statements and re-written a few sentences. It should look more concise now.

#2 The resolution of the graphics is very bad.

Thank you for noticing it. The original figure files have the recommended resolution required by PLOS ONE's. All the original figures file has a resolution of 330 dpi. It may happen when the submission system generates the PDF file. Unfortunately, this is a journal’s problem that we cannot fix.

#3 I would like suggest to insert the figure 1 in the text (line 153)

Thank you for your comment. We included the Fig 1 showing the single-LIG and LIG-chip design.

#4 In the caption of figure 1 I would eliminate the data already reported in the text (Lines from 286 to 288). 

Thank you for your suggestion. We deleted the repeated information from the figure caption. The same information can be found in the main discussion text.

#5 In the “In silico modeling of aptamer structure” (line 271) paragraph, it is not clear why make substitutions in the sequence if you already know that they do not bring advantages. 

Thank you for your questions. We made the T substitutions in the sequence to test the affinity levels of aptamer towards the target and non-target viruses. In addition, scrambling an aptamer sequence is important during in silico modeling to confirm stability of secondary structure, since stability is essential during immobilization step on electrode surface. You can check the statement about stability of secondary structure in the results section. Also, we had the following statement to clarify the modification on aptamer sequence: “In addition, the scrambled aptamer was used as a control to test different binding affinities towards the target analyte and discrimination between potential non-target viruses.”

#6 Are the values obtained from the OCP analysis significant? (is the difference significant?) 

Thank you for your question. We performed a statistical analysis (ANOVA/ Tukey test) to verify if the OCP values were significant. You could see in the Figure 3 that same letters represent groups with no significant difference (p < 0.05). We discussed it in the main text for that section. See example in the following sentence: “Saliva (human and pooled) potential differs significantly (between 80 to 90%, p < 0.05) from potential interferent substances (orange juice, Gatorade, coffee, synthetic urine).”

Reviewer #2

#1 In the manuscript the authors follow the sensor's surface functionalization using the impedance signal variation. Have the authors investigated this procedure using additional methods such as atomic force microscopy (AFM)? Thank you for your question. We haven’t use atomic force microscopy. However, we used confocal imaging as an additional method to demonstrate the surface functionalization and aptamer distribution. We added the results from confocal images in the supplementary material (S5 Fig). In addition, we added the following statement in the Methods section, Sensor biofunctionalization: “Confocal imaging of LIG electrodes before and after biofunctionalization can be found on Supplementary material (S5 Fig).”

#2 In the manuscripts, no information about the reproducibility and the long-term stability of the biosensor developed has been provided by the authors. Please, the authors should provide it. 

Thank you for your suggestion. We had evaluated the reproducibility and shelf-life of the aptasensor. Because the aptasensor will integrate a multiplex sensing platform, with an ACE-based biosensor, we will publish both information in a separated manuscript, which is in preparation.

#3 How authors have determined the sensitivity of the proposed biosensor? 

Thank you for your question. Analytical sensitivity was calculated as the slope of the calibration curve (Fig. 5) in the dynamic concentration range (10 to 1000 copies/ml), where the behavior is approximately linear. We included the following statement about analytical sensitivity calculation in the methods section: “Analytical sensitivity was calculated as the slope of the calibration curve in the dynamic concentration range where the behavior is approximately linear.”

#4 Figure 4° lacks the mathematical equation used by the authors to calculate the biosensor LOD. Please, the authors should provide it. 

Thank you for your comment. You can find the mathematical equation from the non-linear model (MnMolecular1) fitted to obtain the calibration curve and used to estimated LOD in the Supplemental material (S4 table). We have added the equation to the figure.

#5 How have the authors selected the interference substances (coffee, Gatorade, synthetic urine, etc)? What is the pH of these samples? The interferent solutions were diluted 10% v/v in buffer, why have the authors selected this value? Have the authors investigated other concentrations? If yes, what is the matrices effect on the virus binding?

Thank you for your question. We selected interference substances that could mislead the test results since the testing has been designed to autonomous saliva sampling. To guarantee saliva sample integrity, we used a bare LIG electrode to test the potential (V) of human saliva compared with those substances. We did not intend to check the matrices effects on the virus binding, since we did not use a functionalized biosensor for this assay purpose. The substances have different pH, and we had added this information in the supplemental material. The solutions were diluted 10% in buffer because that is the dilution factor, we used for saliva sensing.

To clarify the purpose of this approach the following statement was added to the methodology “Open Circuit Potential (OCP) technique was applied to determine saliva's potential (V) and different interferent substances that could mislead the results when human saliva is not used in the sensing.” In addition, we added a table with substances pH in the supplementary material and added the following sentence to results section: “The pH measurement of all substances can be found in Supplementary material (S4 Table).”

#6 In the title the acronymous “LIG” should be specify. 

Thank you for your comment. We specified the acronymous LIG in the title.

---

## [Editor Report · Decision Letter 1]

4 Aug 2023

A capacitive laser-induced graphene based aptasensor for SARS-CoV-2 detection in human saliva

PONE-D-23-18219R1

Dear Dr. Vanegas,

We’re pleased to inform you that your manuscript has been judged scientifically suitable for publication and will be formally accepted for publication once it meets all outstanding technical requirements.

Kind regards,

Sabato D'Auria

Academic Editor

PLOS ONE
---

## [Editor Report · Acceptance letter]

10 Aug 2023

PONE-D-23-18219R1 

A capacitive laser-induced graphene based aptasensor for SARS-CoV-2 detection in human saliva 

Dear Dr. Vanegas:

I'm pleased to inform you that your manuscript has been deemed suitable for publication in PLOS ONE. Congratulations! Your manuscript is now with our production department. 

Kind regards, 

on behalf of

Dr. Sabato D'Auria 

Academic Editor

PLOS ONE